# Structural evidence for elastic tethers connecting separating chromosomes in crane-fly spermatocytes

Arthur Forer[1] , Shotaro Otsuka[2,3]

**Different types of anaphase bridges are reported to form between segregating chromosomes during cell division. Previous studies using laser microsurgery suggested that elastic tethers connect the telomeres of separating anaphase chromosomes in many animal meiotic and mitotic cells. However, structural evidence is lacking for their existence. In this study, by correlating live imaging with electron tomography, we examined whether visible structures connect separating telomeres in meiosis I of crane-fly primary spermatocytes. We found structures extending between separating telomeres in all stages of anaphase. The structures consist of two components: one is darkly stained, looking somewhat like chromatin, whereas the other is more lightly stained, appearing filamentous. Although in early anaphase both structures extend between telomeres, in later anaphase, the darker structure extends shorter distances from the telomeres but the lighter structure still extends between the separating telomeres. From these observations, we deduced that these structures represent the "tethers" inferred from the laser-cutting experiments. Because elastic tethers have been detected in a variety of animal cells, they probably are present during anaphase in all animal cells.**

## Introduction

The existence of elastic structures that connect the anaphase telomeres of separating partner chromosomes was deduced from phenomenological evidence in crane-fly primary spermatocytes. After a chromosome arm is severed with a laser microbeam in early anaphase, the resultant arm fragment moves rapidly across the equator until it reaches the partner telomere (LaFountain et al, 2002; Forer et al, 2017; Sheykhani et al, 2017). Both telomeres are necessary for arm fragment movement; ablation of either telomere (that of the arm fragment or that of the partner) causes the arm fragment to stop moving (LaFountain et al, 2002; Forer et al, 2021). Furthermore, when the original arm fragment is cut in half, only the

portion with the telomere moves to the partner (LaFountain et al, 2002). Thus, elastic "tethers" were inferred to form between partner telomeres and exert backward forces on anaphase chromosomes. Similar phenomenological evidence showed that all separating chromosomes seem connected by tethers in mitotic and meiotic cells across the animal kingdom (Forer et al, 2017), ranging from turbellarian flatworms, to spiders, to dipteran and orthopteran insects, to marsupials (PtK cells), and to humans (U2OS cells). However, we know of no *ultrastructural* evidence that identifies physical connections between separating telomeres.

Arm-fragment movement is not due to microtubules. When cells are treated with the microtubule-stabilizing agent taxol, anaphase chromosome movements are greatly slowed but the arm fragments move at their usual speeds (Forer et al, 2018). Nor are the movements due to ultrafine DNA bridges extending between separating chromosomes. Although tethers are expected to form between all of the separating telomere pairs (Sheykhani et al, 2017; Forer et al, 2018, 2021; Forer & Berns, 2020), ultrafine DNA bridges are present in far fewer numbers (Barefield & Karlseder, 2012; Gemble et al, 2015, 2016; Nielsen et al, 2015; Kong et al, 2023) (details in the Discussion section). Furthermore, ultrafine DNA bridges slow down anaphase chromosome movements (Su et al, 2016), but tethers do not (Forer et al, 2017; Sheykhani et al, 2017) (details in the Discussion section). Thus, arm-fragment movements seem to be due to a structural, elastic component, not to microtubules and not to ultrafine DNA bridges. We do not know the exact function of tethers but some evidence suggests that tethers are involved in coordinating movements between partner chromosomes (Sheykhani et al, 2017; Forer et al, 2018; Forer & Berns, 2020; Fegeras-Arch et al, 2020).

One should be able to visualize tethers if they are indeed structures that connect separating anaphase telomeres. Light microscopically, studies of cell division in fixed/stained cells in the early- to mid-1900s illustrated connections between separating chromosomes (Paliulis & Forer, 2018), as have more recent studies (Bajer & Mole-Bajer, 1986). Although inter-telomere connections were illustrated in these earlier studies, the authors often did not comment on them because the studies concerned mostly spindle fibres and how chromosomes move. In articles in which they were mentioned, there was not sufficient description to be sure the

[1]Biology Department, York University, North York, Canada    [2]Max Perutz Labs, Vienna Biocenter Campus, Vienna, Austria    [3]Medical University of Vienna, Center for Medical Biochemistry, Vienna, Austria

Correspondence: aforer@yorku.ca; shotaro.otsuka@univie.ac.at

images fit what we know about tethers. Thus, we know no light microscopic method that for sure visualizes tethers. Electron microscopically, most studies of mitotic and meiotic spindles have concentrated on spindle microtubules. We know of only two articles in which structures connecting separating telomeres have been described (discussed in Forer et al [2017]). One, by Fuge (1978), describes physical connections between separating half-bivalents in two fortuitously sectioned anaphase crane-fly spermatocytes. Fuge considered the connections as extended chromatin. The other, by Krishan and Buck (1965), described, connections between the tapered ends of separating chromosomes in anaphase cockroach spermatocytes. Krishan and Buck thought these connections might represent coated microtubules. These two articles were one-time observations, and not considered general. We thus know of no articles that satisfactorily describe the consistent presence of presumptive tethers; however, we know of none that deny their existence, either.

In this article, we present electron microscopic evidence of physical connections between telomeres of separating anaphase chromosomes that were seen in all six anaphase crane-fly spermatocytes that we observed. Although we cannot prove these structures are the "tethers" defined in experiments that followed movements of severed chromosome arms, they are not microtubules and they satisfy what we know about tethers. We thus suggest that these structures represent the tethers inferred from the laser-cutting experiments in living cells (LaFountain et al, 2002; Forer et al, 2017, 2021).

# Results

### Tethers connect separating anaphase chromosomes

The existence of elastic structures that connect the telomeres of separating chromosomes from early anaphase to telophase was suggested by previous cell biological studies. To examine if such elastic structures actually exist, we studied the ultrastructure of separating anaphase chromosomes using 3D electron microscopy. Living primary spermatocytes in meiosis I isolated from crane-fly larvae were initially observed by light microscopy. Anaphase cells were identified and then fixed with glutaraldehyde, dehydrated, and embedded in resin (see the Materials and Methods section). We sectioned the resin-embedded cells at a thickness of 250 nm and visualized their 3D ultrastructure by electron tomography. The experimental workflow is illustrated for one mid-to-late anaphase cell in Fig 1A. Overall, the resin-embedded cells looked like the live cells (Figs 1A and B and S1A). In the sectioned cells, there were very few vacuoles, mitochondria appeared to be fixed properly, and microtubules looked normal (Figs 1C, 2B, and S1B), indicating that there was a good structural preservation. Intriguingly, we found filamentous structures, clearly morphologically distinct from microtubules that extended between telomeres on separating anaphase chromosomes (Figs 1C and D, 2B, and S2 and Video 1). These presumed "tethers" were observed in all the six different anaphase cells that we sectioned, which were in stages from very early anaphase to mid-to-late anaphase. The images shown in Figs 1–3 are examples of sections in which we could see, and illustrate, the

entire length of the filaments connecting partner telomeres. The telomeres were identified as the termini of the trailing arms. Some of the tethers were tilted against the section plane and appeared in two consecutive sections (Figs 2A and 4A). These tethers in cells with extensions from only one telomere, looked like the tethers that extended between partner telomeres, and thus we included them in the morphology measurements. Our analysis of visible filamentous connections between telomeres includes 15 tethers, from six cells, as indicated in Table 1.

### Tethers have distinct substructures

The tethers that connect the telomeres seem to have two components. One is darker (more electron dense), with irregularly shaped, darkly stained internal structure, looking somewhat like chromatin (Figs 1C, 2B, 3B and D, and 4B and Video 1). The other is more lightly stained (Figs 1C, 2B, 3D, and 4B and Video 1), appearing filamentous with internal thinner filaments arranged more-or-less linearly parallel to each other (Fig 2B). In early anaphase, the tethers that arise are primarily dark and look like pulled-out extensions of chromatin (Fig 3A–F); the lightly stained tethers also have chromatin-like extensions at the telomeres (Fig 3B and D).

The darkly stained, pulled-out tether regions at the telomeres change as tethers elongate. In short tethers, the pulled out dark region is thick and tapers to a point at a steep angle (Fig 3B [ii] and Fig 3D [i]) compared with longer tethers in which the pulled out dark region is thinner and more sharply pointed (Figs 2B, 4A and B). Quantitative analysis of the neck and midline width revealed that the extensions from telomeres in early anaphase are thicker at the neck than extensions seen later in anaphase, changing from 90–180 nm in early anaphase to 50–90 nm at mid/late anaphase (Fig 3G). On the other hand, the midline width of the extensions was similar between early anaphase and mid/late anaphase, ranging from 35 to 55 nm (Fig 3G). In a very early anaphase cell, most of the extensions were wider than 250 nm (the thickness of the sections) (Fig 3E and F). When the dark and light components are both present, the dark material seems to be on one side of the lighter material (Figs 1C, 2B, 3B and D, and 4B). The darker (more electron dense) tethers are statistically significantly wider than the lighter tethers (the average midline width was 46 and 33 nm for the darker [n = 13] and lighter [n = 14] tethers, respectively (Fig 3G). Interestingly, ultrastructural inspection of the lighter tethers demonstrated that they seem to consist of bundles of thin filaments, whose width was around 5 nm (Fig 2B).

In sum, the tethers have darker and lighter components. The darker components remain only near the tip of the telomere in mid/late anaphase, whereas the lighter components still connect two chromosomes in mid/late anaphase. Nearer to the telomeres, the darker component is on the outside of the lighter filamentous component.

# Discussion

Our main conclusion is that filamentous structures connect partner telomeres as the chromosomes move to opposite poles during anaphase in crane-fly primary spermatocytes. We think that these

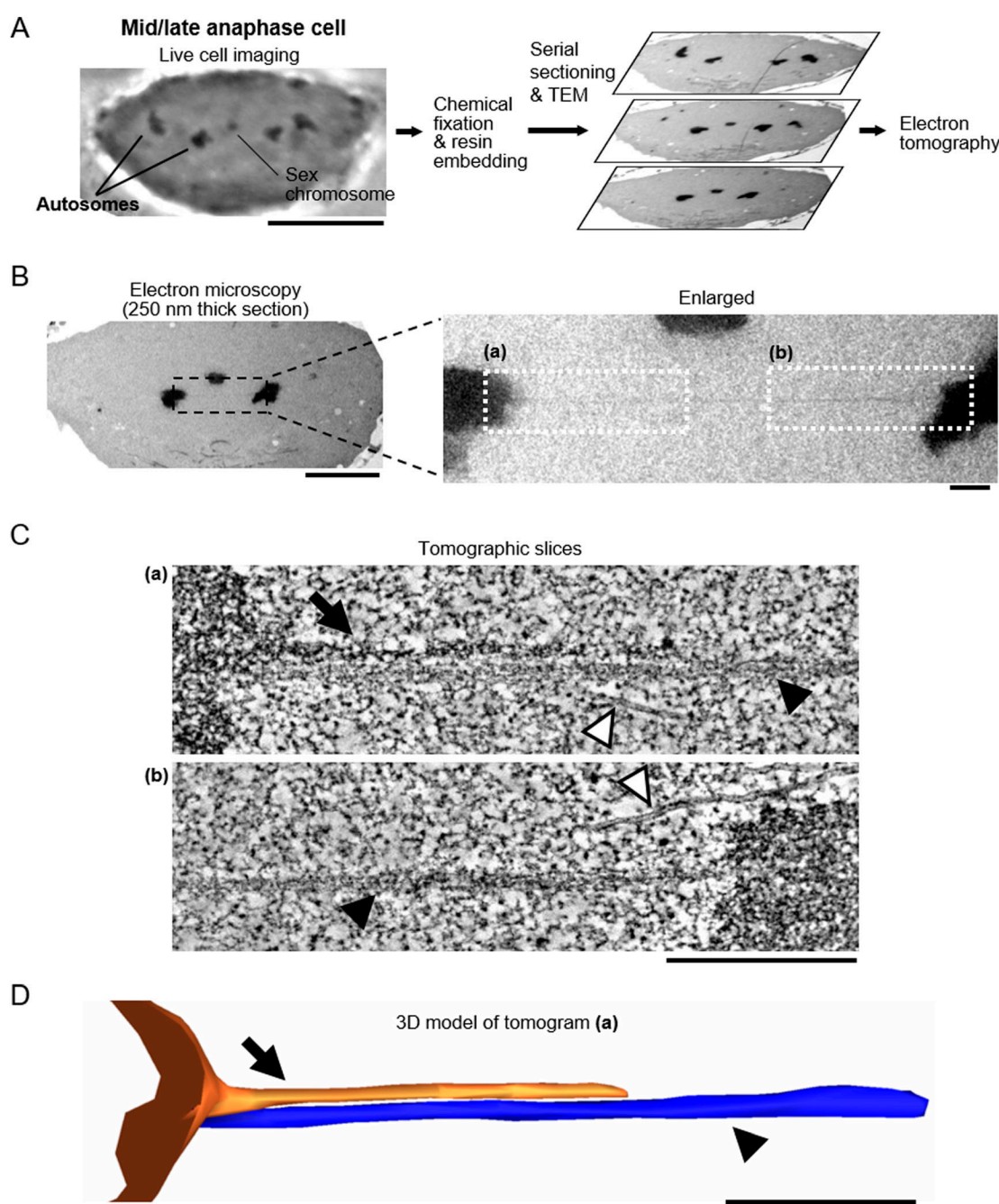

**Figure 1. Filamentous structures are forming between separating chromosomes of a mid/late anaphase cell.**
**(A)** The experimental workflow. The cell division of a crane-fly spermatocyte is monitored by light microscopy (left). Two of the autosomes and one sex chromosome are indicated. The anaphase cell was instantly fixed with glutaraldehyde. After being embedded in plastic resin, the same cell was relocated in the resin block. The serial sections were cut at a thickness of 250 nm and observed by transmission electron microscopy (right). The 3D structure of the cell in each section was investigated by electron tomography. Scale bar: 10 μm. **(B)** Electron microscopy images of one of the sections of the mid/late anaphase cell shown in (A). The regions that were analyzed by electron tomography are indicated in white boxes in the enlarged image. Scale bars: 5 μm (left) and 500 nm (right). **(C)** Electron tomography analysis of the filamentous structures indicated in (B). Projections of 30 tomographic slices (corresponding to 17 nm thickness) are shown. White arrowhead: microtubule; black arrow: darker tether; black arrowhead: lighter tether. Scale bar: 500 nm. **(D)** 3D mesh of the darker (orange) and lighter (blue) tethers shown in (C) (a). The left half of the tethers is highlighted. Black arrow: darker tether; black arrowhead: lighter tether. Scale bar: 500 nm. See also Video 1.

structures are the "tethers" that were deduced to be present from experiments in which lasers cut arms from anaphase chromosomes, after which the arm fragments moved backwards to their partner chromosome, telomere moving to telomere (LaFountain et al, 2002; Forer et al, 2017, 2021). We have no direct proof that the structures that we identified using electron tomography are the tethers deduced from laser experiments. However, the fact that the structures extended over 5 μm between telomeres on separating

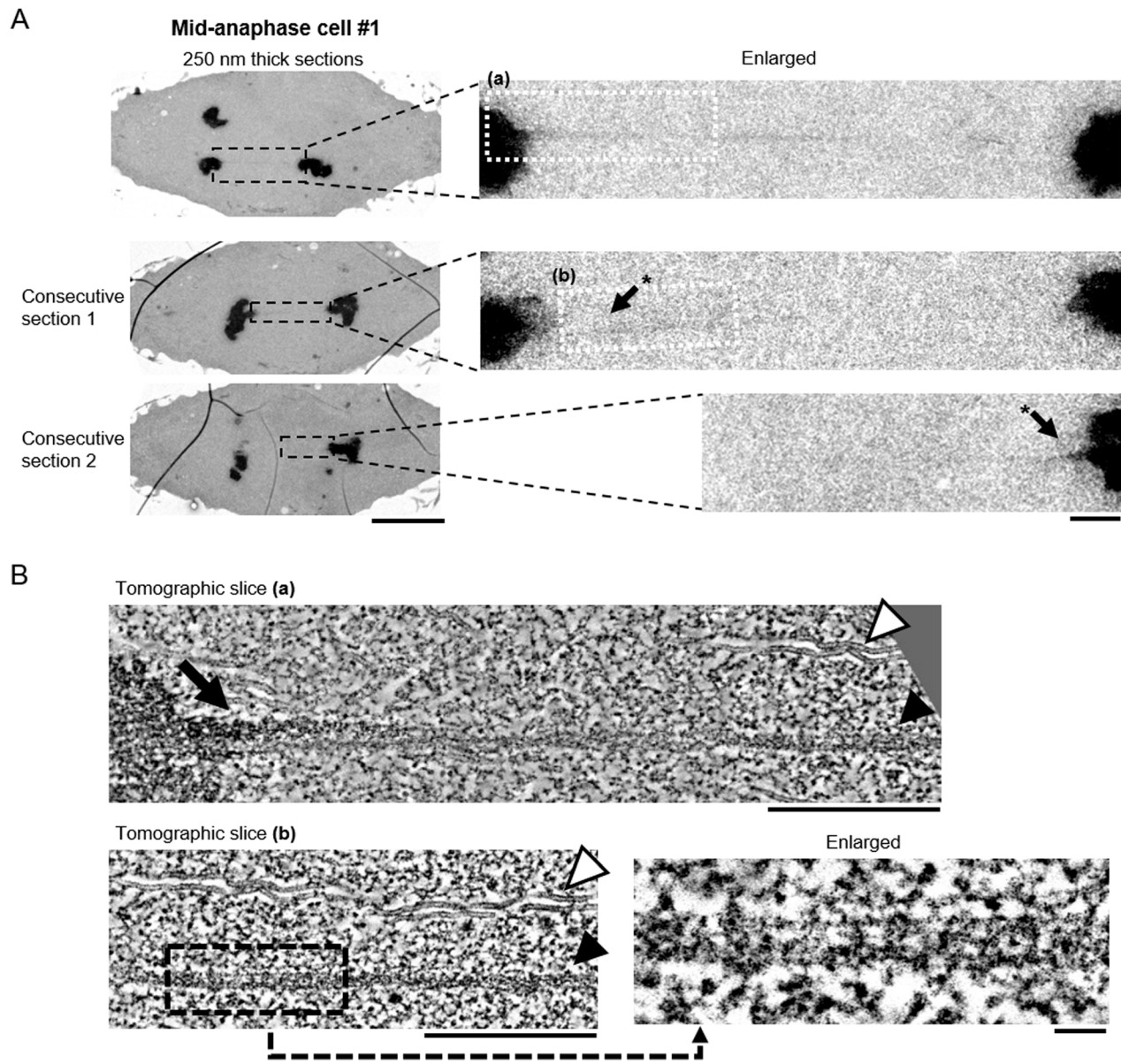

**Figure 2. Filamentous structures extending between separating chromosomes are also seen in a mid-anaphase cell.**
**(A)** Electron microscopy images of three sections of a mid-anaphase cell. The regions that were analyzed by electron tomography are indicated in white boxes in the enlarged images. Scale bars: 5 μm (left) and 500 nm (right). *The same tether appeared in the two consecutive sections. **(B)** Electron tomography analysis of the filamentous structures indicated in (A) (a, b). Projections of 30 tomographic slices (corresponding to 17 nm thickness) are shown. For the tomographic slice (b), enlargement of a region indicated in a black box is shown to highlight the sub-filaments. White arrowhead: microtubule; black arrow: darker tether; black arrowhead: lighter tether. Scale bars: 500 nm (top and left) and 50 nm (right bottom).

anaphase chromosomes, and the fact that we did not find any other connecting structures in our chemically-fixed cells (Figs 1B and 2A and Table 1), strongly suggests that they are the tethers.

Electron microscopically visible connections between separating anaphase chromosomes have been illustrated in a few other cells. In one, Krishan and Buck (1965) described telomere-to-telomere connections in early anaphase cockroach spermatocytes. Their images (Krishan & Buck, 1965) are not dissimilar from ours, as are some of their descriptions (anaphase chromosome "equatorial ends become drawn out into narrow tapering fingers"). In another cell type, Fuge (1978) described telomere-to-telomere connections in two anaphase spermatocytes from a species of crane fly (*Nephrotoma ferruginea*) different from the one we studied. Fuge argued that the fact that connections between telomeres (which he called "bridges") were "observed in three out of six bivalents (chromosomal set 2n = 6 autosomes + XY) suggests the phenomenon to be a rather frequent one…" He measured the widths of "bridges" ("tethers") as 41–46 nm, close to the tether widths we measured (46 and 33 nm for the darker and lighter tethers, respectively (Fig 3G). He further described the "bridges" (tethers) as

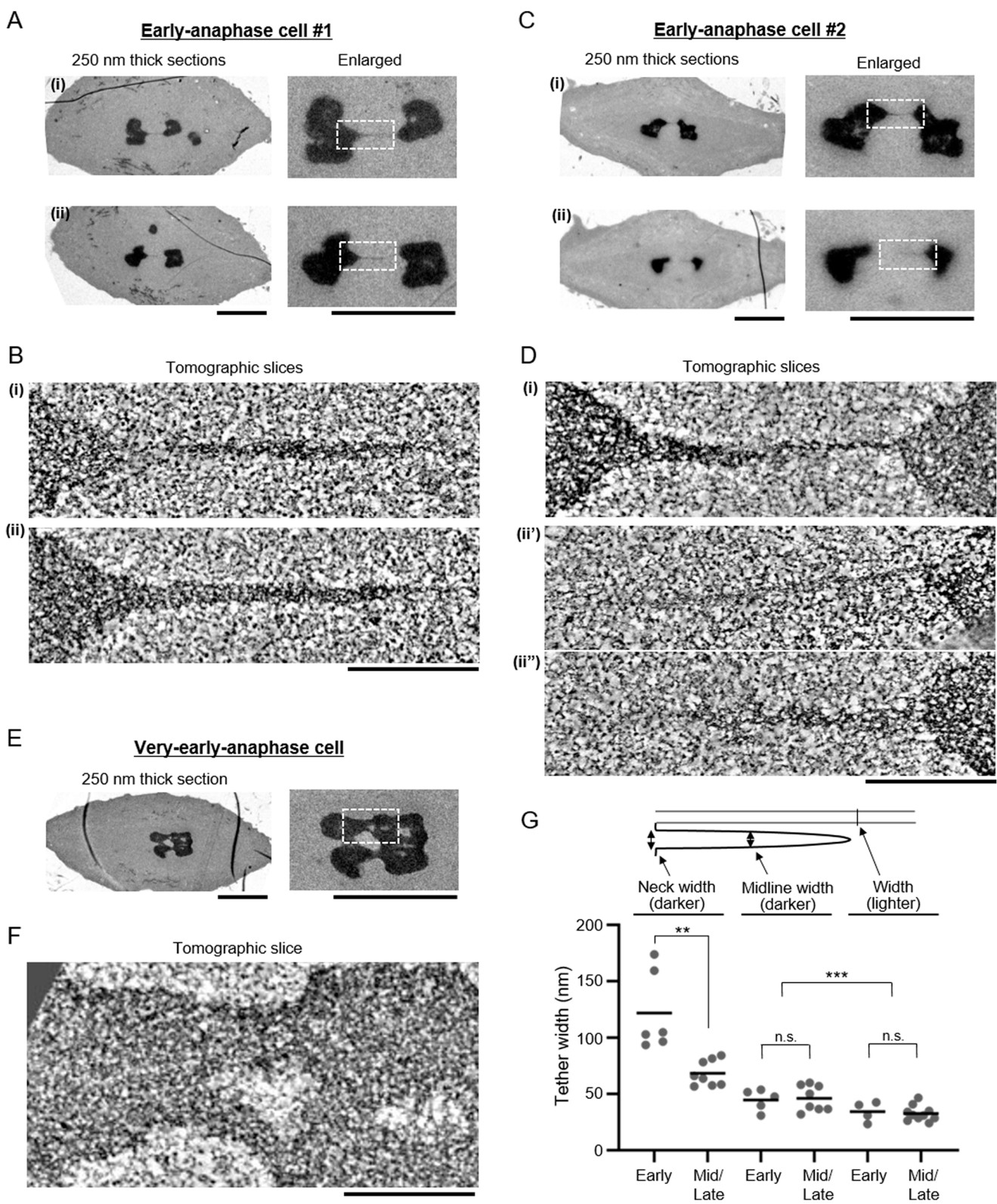

**Figure 3. The filamentous structures are thicker in early anaphase cells.**
**(A, C, E)** Electron microscopy images of two sections of early anaphase cells (A, C), and a section of a very early anaphase cell (E). The regions that were analyzed by electron tomography are indicated in white boxes in the enlarged images. Scale bar: 5 µm. **(B, D, F)** Electron tomography analysis of the filamentous structures indicated in (A, C, E). Projections of 30 tomographic slices (corresponding to 17 nm thickness) are shown. Scale bars: 500 nm. (ii') (ii''): different slices in the same tomogram (D). Scale bar: 500 nm. **(G)** Quantification of the tether width as indicated by the illustration. The tethers that we measured are from two early anaphase cells (labelled as "Early")

having filamentous substructures of around 10 nm in width. We too saw thin filaments composing the structure of the lighter stained tethers (Fig 2B), which, in our sections, were around 5 nm thick. In a third cell type, Fawcett (1981) illustrated electron dense chromatin-like connections between separating arms in early anaphase of a Chinese hamster spermatogonium cell Fawcett (1981), images reminiscent of those we see in early anaphase in crane-fly spermatocytes (Fig 3). The fact that in electron microscope tomograms we regularly see connections between separating anaphase telomeres in crane-fly spermatocytes, together with these sporadic sightings in cockroach spermatocytes, in Chinese hamster spermatogonia, and in spermatocytes of a different crane-fly species, suggest that similar structural tethers between separating anaphase telomeres are a regular occurrence. They probably have not been seen until recently because of the difficulty in seeing such thin components without being exactly in the plane of the tethers, and because most electron microscopy has concentrated on the spindle proper and not on the interzonal region between separating chromosomes in anaphase.

How are tethers formed? The extended tether has two components: a darker component with amorphous chromatin–like staining, and a more filamentous component, stained less darkly. The darker component remains near the telomere as the tethers become longer; for lighter tethers, most of the tether length appears filamentous. Where the two elements co-exist in the sections, the darker component is on the outside of the lighter filamentous component. We can suggest two possibilities of how tethers form. One possibility is that the filamentous component pre-exists in the chromosome arm and as it is pulled out from the telomere region it also pulls out its covering sheath (the darkly stained material). That would be why the chromatin-like tip of the telomere remains pulled out when tethers are longer (Figs 2 and 4). However, we cannot rule out another possibility. It might be that only the darkly stained amorphous chromatin-like component is pulled out. The same material is rearranged into filamentous components as the linear force induces the elastic components to line up, perhaps as a rumpled up bunch of rubber bands would align into linear elements as it is stretched in one direction.

What are tethers made of? We do not know. But whatever the elastic component is, tethers generally are elastic at shorter tether lengths (<5 $\mu$m) and gradually become more inelastic at longer lengths (>7 $\mu$m) and are completely inelastic when tether lengths are >10 $\mu$m (LaFountain et al, 2002; Forer et al, 2017, 2021; Kite & Forer, 2020). This is deduced from studying moving arm fragments that are severed from the tips of chromosome arms in crane-fly spermatocytes and PtK cells; arm fragments move but often do not reach the other telomere when tether lengths are >5 $\mu$m, but arm fragments do not move at all when tether lengths are >10 $\mu$m. This is not because the elasticity is stretched past a breaking point, like a bungee cord might, or because the tether detaches from the telomeres. This is rather because the elasticity of the tether is controlled by phosphorylation; blocking protein phosphatase 1

(PP1) in early anaphase causes the tethers to remain elastic throughout anaphase and into telophase (Fabian et al, 2007a; Kite & Forer, 2020; Forer et al, 2021). Fabian et al (2007b) stained crane-fly spermatocytes with antibodies against titin, the prominent muscle protein, and found titin extending between telomeres of separating anaphase telomeres, apparently connecting them. Fabian et al (2007a, 2007b) suggested that titin was the elastic component of the tethers originally identified by LaFountain et al (2002). Subsequent experiments showed that tether elasticity is controlled by phosphorylation, with PP1 causing loss of phosphorylation and loss of tether elasticity (Kite & Forer, 2020). This is consistent with the suggestions that tethers contain titin because the elasticity of titin is also influenced by phosphorylation; titin loses elasticity when it is dephosphorylated by PP1 (Krüger & Linke, 2006; Hidalgo & Granzier, 2013; Kötter et al, 2013; Hamdani et al, 2017), just as tethers are. As far as we know, there have been no other suggestions as to what tethers might be made of.

Are there any other components that connect anaphase chromosomes that might be represented by the structures we described? Ultrafine DNA bridges act as connecting "bridges" between separating chromosomes, but they are different from what we know about elastic tethers and what we identified electron microscopically as elastic tethers. Ultrafine DNA strands arise from centromeres, from fragile sites, from ribosomal DNA, and from telomeres, in differing numbers (Gemble et al, 2015, 2016; Kong et al, 2023). The vast majority of ultrafine DNA bridges connect centromeres on separating anaphase chromosomes (Nielsen et al, 2015; Kong et al, 2023). Such ultrafine DNA strands/bridges are not the tethers we observed because most of the ultrafine DNA bridges are between centromeres. Even if all of them were between telomeres, they would still be present in far too few numbers to be elastic tethers. In crane-fly spermatocytes, each separating anaphase chromosome pair is connected by two tethers out of the four separating arms (LaFountain et al, 2002; Sheykhani et al, 2017), so in this meiotic cell one expects to find two connections per chromosome pair, or six per cell. In anaphase PtK cells, at least 75% of the separating chromosomes are connected by tethers, and most likely 100% are (Forer et al, 2017), so in looking for connections between separating anaphase telomeres one would require structural connections corresponding to at least one bridge per separating chromosome pair. Ultrafine DNA bridges are present in human cells with average frequencies of 1–6 bridges per cell, out of 46 chromosomes in diploid cells (Gemble et al, 2015, 2016; Nielsen et al, 2015; Kong et al, 2023), and at least 20% of cells studied have zero bridges (Barefield & Karlseder, 2012; Gemble, et al, 2015; Nielsen et al, 2015; Kong et al, 2023). Therefore, the numbers of ultrafine DNA bridges connecting telomeres are far too low to represent tethers identified by the laser experiments. On the other hand, electron microscopically we have seen up to six tethers per primary spermatocyte, which is the predicted number of tethers in meiosis with 2n = 6 autosomes. Another reason that tethers identified

and one mid/late anaphase cell plus two mid-anaphase cells (labelled as "Mid/Late"). The mean is depicted as a horizontal line. **$P$ < 0.01, ***$P$ < 0.001; unpaired $t$ tests. n.s., not significant.
Source data are available for this figure.

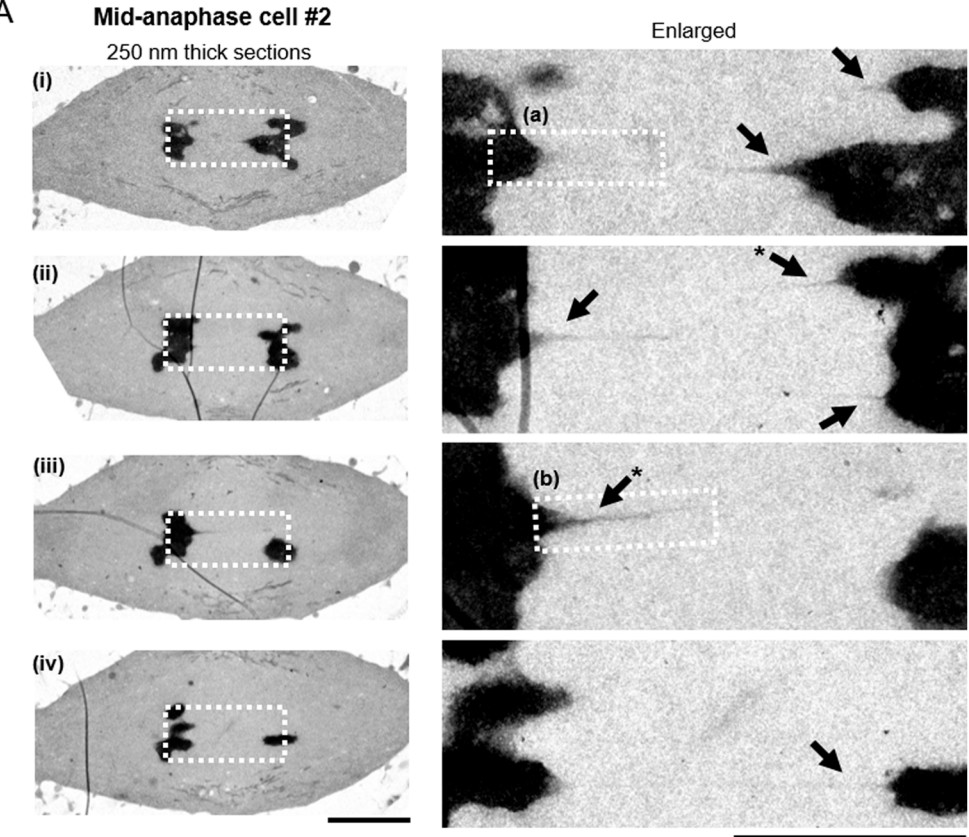

**Figure 4. The filamentous structures are observed regularly in another mid-anaphase cell.**
**(A)** Electron microscopy images of four sections of another mid-anaphase cell. The regions that were analyzed by electron tomography are indicated in white boxes in the enlarged image. Scale bars: 5 µm (left) and 500 nm (right). Black arrows: darker tethers. *The same tether appeared in the two consecutive sections. **(B)** Electron tomography analysis of the filamentous structures indicated in (A). Projections of 30 tomographic slices (corresponding to 17 nm thickness) are shown. Black arrow: darker tether; black arrowhead: lighter tether. Scale bar: 500 nm.

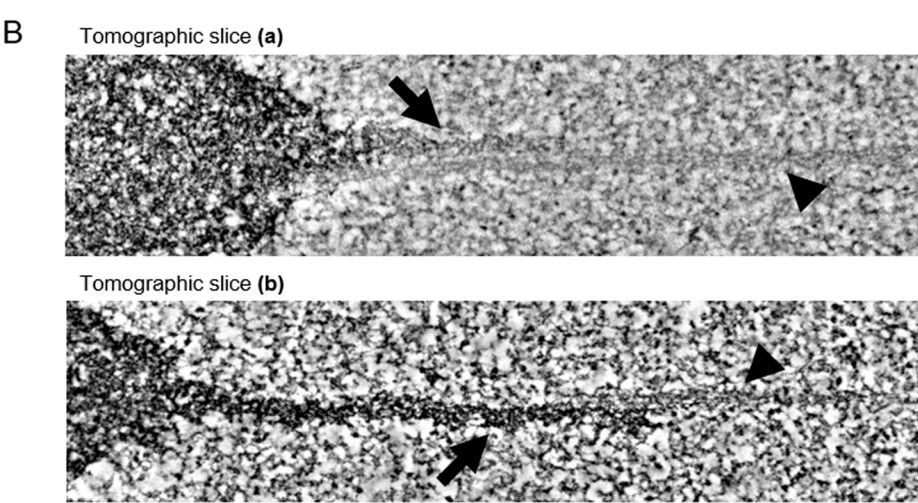

using laser cuttings are not ultrafine DNA bridges is that properties of ultrafine DNA bridges do not match tether properties. When one experimentally increases the numbers of ultrafine DNA bridges, the anaphase chromosomes slow down (Su et al, 2016), whereas tethers do not cause chromosomes to slow down, even when the tethers have lost their elastic contractility (Forer et al, 2017; Sheykhani et al, 2017; Forer & Berns, 2020; Kite & Forer, 2020). Thus, the

images we obtained correspond to what one expects of tethers, not of ultrafine DNA strands.

Connections between separating anaphase chromosomes were described in *Drosophila* neuroblasts that seem to be different both from ultrafine DNA bridges and from elastic tethers (Warecki et al, 2023). The usual ultrafine DNA bridges do not stain with standard DNA dyes such as DAPI and do not contain histone

**Table 1. Overview of the tethers.**

| Stage | Number of tethers studied | Distance between telomeres ($\mu m$) |
| --- | --- | --- |
| Mid–late anaphase cell (Fig 1) | 1 | 5.3 |
| Mid-anaphase cell 1 (Fig 2) | 2 | 5.1–5.6 |
| Early anaphase cell 1 (Fig 3A and B)[a] | 2 | 1.5–1.7 |
| Early anaphase cell 2 (Fig 3C and D) | 4 | 1.0–2.5 |
| Very early anaphase cell (Fig 3E and F) | n.d. | 0.48 |
| Mid-anaphase cell 2 (Fig 4) | 6 | 3.5–5.0 |

The number of tethers studied and the distances between telomeres are presented in six different anaphase cells.
[a]For early anaphase cell 1, only half of the cell was sectioned, and thus it is possible that it contains more tethers.

(Ke et al, 2011; Gemble et al, 2015, 2020; Nielsen et al, 2015; Kong et al, 2023); they rather are identified by staining with associated proteins such as PICH (Nielsen et al, 2015). The bridges in *Drosophila* neuroblasts described by Warecki et al (2023) were identified as containing DNA by staining with DAPI, and some were recognized as containing histones (thought to arise from recruitment of histone to DNA bridges), so they do not seem to be ultrafine DNA strands. Nor do they seem to be the elastic tethers we have described because there are too few per cell. *Drosophila* has 2n = 6 autosomes, so one would expect to regularly identify six DAPI-positive bridges in each cell if these structures were tethers and if tethers contained DNA. Warecki et al (2023) saw anaphase "bridges" between chromosomes, primarily at telomeres, but saw them in only 38% of the cells, and mostly in early anaphase. Thus, these are not the same as ultrafine DNA bridges, or the same as the tethers we described, or that were described by the laser experiments; if their results are found to be generally applicable, the bridges would seem to be yet another (third) kind of "tether" or "bridge."

In summary, we have described structural filamentous connections that appear regularly between separating anaphase telomeres in crane-fly spermatocytes. We have presented evidence that these filaments represent the elastic tethers between telomeres originally described by LaFountain et al (2002). Because similar elastic tethers are present in a variety of animal cells, from freshwater flatworms to humans (Forer et al, 2017), we expect such structures to be commonly seen in other cell types as well if they are examined at high spatial resolution, ideally by using 3D electron microscopy techniques such as electron tomography that allows visualization of intracellular structures at an isotropic spatial resolution below a few nanometers in 3D.

# Materials and Methods

## Living material

We studied spermatocytes from crane flies (*Nephrotoma suturalis* Loew) that were prepared in the laboratory using methods similar to those described in Forer (1982). In brief, pupae were placed on moist *papier mâché* in a cage. The adults that emerged mated and laid eggs on fresh moist *papier mâché*. After the eggs hatched, the resultant larvae were kept on *papier mâché* in Petri dishes and fed

with powdered nettle leaves twice weekly until the larvae molted to become pupae. The pupae were placed in a cage to start the cycle again. We obtained spermatocytes from fourth-instar larvae. The cells in crane-fly testes are more-or-less synchronous and there is a roughly 2-d period during the final (fourth) instar in which most cells in the testis undergo meiosis I. We chose animals to dissect as near the peak period as we could (described in Forer [1982]). We dissected testes under a mixture of heavy and light halocarbon oil (Sigma-Aldrich), and spread the cells on coverslips in insect Ringer's solution (0.13 M NaCl, 5 mM KCl, 1.5 mM CaCl$_2$, 3 mM KH$_2$PO/Na$_2$HPO$_4$ buffer, final pH 6.8) that contained fibrinogen (procedures described in detail in Forer and Pickett-Heaps [2005]). We added thrombin to form a clot that embedded the cells, and immersed the cells in insect Ringer's solution (Forer & Pickett-Heaps, 2005). In the present experiments, the cells were placed in the 16 mm diameter central chamber of black-anodized aluminum slides (Forer & Pickett-Heaps, 2005). We observed the cells using a Nikon Diaphot microscope with a 40x NA 1.3 or 100x NA 1.3 phase-contrast objective. We scanned the cells, noted that they looked healthy, and when a suitable living anaphase cell was seen, we removed the coverslip from the chamber and within seconds fixed the cells with 2.5% glutaraldehyde. The 2.5% glutaraldehyde was prepared from stock 25% glutaraldehyde (JB EM Services) by diluting the stock glutaraldehyde by a factor of 10 with insect Ringer's solution. All cells studied were fixed within 20 min of making the living cell preparations. When we saw no cells in anaphase in the living cell preparation within 20 min, we discarded the slide and made a new preparation. For fixation, we put a large drop of glutaraldehyde onto the bottom of a plastic 60 mm diameter Petri dish and put the coverslip and attached cells on top of the drop, cells side down.

## Preparations for electron microscopy

The glutaraldehyde fixation was sometimes carried out with room temperature (RT) glutaraldehyde, but most of the cells described in this article were fixed with glutaraldehyde that was kept at 4–6°C until it was added to the cells. The tethers we identified were seen when cells were fixed with either glutaraldehyde at 4–6°C or at RT (Fig S2). During the fixation, the cells and glutaraldehyde were kept at RT. The purpose of the brief cold shock during initial fixation was to try to reduce the number of interzonal microtubules. The cells remained in 2.5% glutaraldehyde (at RT, with no refrigeration) for at

least 30 min. For some preparations, the fixation/staining procedure continued the same day the preparations were fixed with glutaraldehyde. Other preparations were kept in glutaraldehyde and placed in a refrigerator at 4–6°C overnight and the procedures continued the next day. For the preparations illustrated in this article, the cells were fixed with osmium tetroxide the same day as the initial glutaraldehyde fixation but then were stored overnight in 70% ethanol before the embedding was finished on the next day. Continuing the details of the procedure: the preparations in glutaraldehyde were rinsed twice with PBS, the cells were treated with 1% OsO$_4$ in H$_2$O (Electron Microscopy Sciences) for 30 min, rinsed with H$_2$O, treated with 1% uranyl acetate (Electron Microscopy Sciences) in H$_2$O for 30 min, rinsed with H$_2$O, and then stored in 70% EtOH at 4–6°C until all the preparations were ready. After samples were accumulated, the procedure resumed: preparations were dehydrated (at RT) through a series of acetone concentrations, from 70% to 100%. Then the preparations were placed in 100% acetone twice, after which the cells were impregnated by immersion in one part Epon (Embed 812; Electron Microscopy Sciences): two parts acetone, followed by two parts Epon: one part acetone, and then 100% Epon, twice, for 30 min each. The coverslips were then put in an oven at 60°C for 2 d to harden the Epon.

### Relocating cells on EM grids

Cells to be sectioned were selected in the flat embedments. The blocks were viewed through the coverslip (which still adhered to the Epon). Anaphase cells were located and the positions were marked (with marking pen) on the opposite side of the block, and the cells and surroundings were filmed. Then, the glass coverslip was removed by floating the block, coverslip side down, on 48% hydrofluoric acid until the glass was dissolved. The embedment then was rinsed thoroughly with water and placed at 60°C for at least an hour. The cells were then located, filmed, and their positions marked on the block face with a marking pen to locate them for sectioning. The resin blocks were trimmed according to the marks and the trimmed resin that contained the cells of interest were cut with an ultramicrotome (Ultracut UCT; Leica) at a thickness of 250 nm. The sections were collected on copper–palladium slot grids (Science Services) coated with 1% Formvar (Plano).

### Electron tomography

Gold beads (15 nm) conjugated with protein A (Cytodiagnostics) were absorbed on both sides of the sections as fiducial markers for tomography reconstruction. The sections were post-stained with 2% UA in 70% methanol at RT for 7 min, followed by 3% lead citrate in water (Delta Microscopies) at RT for 5 min to enhance sample contrast. Single or dual axis tilt series were acquired with a Tecnai F20 transmission EM (200 kV; FEI) by using Serial EM software (Mastronarde, 2005). Images were recorded over a –60° to +60° tilt range with an angular increment 1° at a final pixel size of typically 0.6 or 1.0 nm. Tomograms were reconstructed using the R-weighted back projection method implemented in the IMOD software package (Kremer et al, 1996). Dual axis tilt series were aligned using gold fiducial markers whereas single axis tilt series were aligned by patch tracking. Meshes of the junctions were generated using IMOD software after manually tracing the outline of the tethers. The width of the tethers was measured manually in the EM tomograms generated using IMOD software.

## Supplementary Information

## Acknowledgements

This project was supported by laboratory startup funding from the Medical University of Vienna to S Otsuka and by the Vienna Science and Technology Fund (WWTF; project LS19-001) to S Otsuka. The authors acknowledge the electron microscopy facilities at the Vienna BioCenter for technical support. In addition, we thank the members of the laboratories of Shotaro Otsuka (especially Helena Bragulat-Teixidor and Clara-Anna Wagner) for feedback and for generating 3D meshes. The work by A Forer was supported by a Discovery Grant from the Natural Sciences and Engineering Research Council of Canada (RGPIN-2019-06299).

### Author Contributions

A Forer and S Otsuka: conceptualization, resources, data curation, software, formal analysis, supervision, funding acquisition, validation, investigation, visualization, methodology, project administration, and writing—original draft, review, and editing.

### Conflict of Interest Statement

The authors declare that they have no conflict of interest.

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
