## [Reviewer comments · Life Science Alliance]

Life Science Alliance

Structural evidence for elastic tethers connecting separating chromosomes in crane-fly spermatocytes

Arthur Forer and Shotaro Otsuka

DOI: <https://doi.org/10.26508/lsa.02302303>

Corresponding author(s): Shotaro Otsuka, Max Perutz Labs

Review Timeline:

Submission Date:	2023-08-03
Editorial Decision:	2023-08-04
Revision Received:	2023-08-06
Accepted:	2023-08-07

Transaction Report:

Please note that the manuscript was reviewed at *Review Commons* and these reports were taken into account in the decision-making process at *Life Science Alliance*.

Review
COMMONS

Reviews

Review #1

****Summary:**** Forer and Otsuka provide first-rate evidence for tethers fixed in place between separating anaphase chromosomes using electron tomography. The authors traced the anaphase movement of a number of living cells before fixation for examination using electron tomography. The manuscript is clearly written and provides an excellent introduction and discussion of the known literature. The reader will have an excellent background to see the importance of this work.

****Major comments:****

- Are the claims and the conclusions supported by the data or do they require additional experiments or analyses to support them?

No further experiments are needed. The data are very supportive, and extremely clear.

- Are the data and the methods presented in such a way that they can be reproduced? Yes.
- Are the experiments adequately replicated and statistical analysis adequate? Yes.

****Minor comments:****

- Are prior studies referenced appropriately? Yes.
- Are the text and figures clear and accurate? Absolutely.
- Do you have suggestions that would help the authors improve the presentation of their data and conclusions?

The authors are to be congratulated on their major contribution to this study on tethers between separated daughter chromosomes. It is a *tour de force* to go from the living cells to fixing and identifying the same separated chromosomes using electron tomography to see the ultrastructure of the fibers seen *in vivo*.

****Referees cross-commenting****

Thank you reviewer #2. The manuscript should be published. It is an excellent contribution.

Provide contextual information to readers (editors and researchers) about the novelty of the study, its value for the field and the communities that might be interested.

This manuscript is the first to use electron tomography to identify the tethers between separated anaphase chromosomes. Forer and the late Michael Berns and their co-authors have published a number of papers using phase microscopy and lasers to report on the physical nature and elastic properties of these fibers in the past. Forer and Otsuka have presented first-rate evidence for the reality of these structures using electron tomography. This manuscript should be highlighted in the published journal.

The chemical identity of these fibers as the authors state is unclear.

The following aspects are important:

- ***Audience:** describe the type of audience ("specialized", "broad", "basic research", "translational/clinical", etc...) that will be interested or influenced by this research; how will this research be used by others; will it be of interest beyond the specific field?

This exciting contribution will be read by anyone interested in mitosis. It will be of interest to all Cell Biologists because of the careful manner in which the living cells were studied before they were fixed for examination using electron tomography. The readers will be dreaming how they can use this process on their Cell Biology problems.

- Please define your field of expertise with a few keywords to help the authors contextualize your point of view. Indicate if there are any parts of the paper that you do not have sufficient expertise to evaluate.

I am a cell Biologist who has made contributions, both in light microscopy and in transmission microscopy on diving cells, both in tissue culture and *in situ* in avian and zebrafish embryos.

Review #2

In this paper, the authors use light microscopy and electron tomography to study anaphase chromosomes in crane fly spermatocytes. They find that there are two "tether" structures that connect telomeres of sister chromatids. One tether is thicker (denser) and extends between sister chromatids during early but not late anaphase, whereas a second, less-dense tether maintains contact with both sister chromatids in all examined stages of anaphase. The paper makes arguments as to what the tethers could or could not be. Specifically, they are too numerous to be ultrafine DNA bridges seen in various normal or abnormal segregation events and they also do not affect anaphase chromosome motion the same way ultrafine DNA bridges do.

****Major comments:****

The major claim that there are tethers that connect sister chromatids in anaphase is supported by the data. Moreover, the data resolves two types of tethers on the basis of their density. While it is unclear what the composition of the tethers are, the paper makes a convincing case that they cannot be the DNA ultrafine bridges seen in other studies. The discussion has sufficient caveats that most readers will see that more work is needed to identify the composition of the two tethers. In my opinion, no further experiments are needed to support the modest claims of this paper. Therefore, I only have minor comments that may hopefully improve the paper's clarity.

****Minor comments:****

It was argued that the tethers reported here were also seen in other species and cellular contexts, where the imaging work was done with projection EM imaging. Presumably, what is new here is the usage of electron tomography. It would help readers if the authors explained why the electron tomography done here was essential to arrive at key conclusions.

p.3 mitochondria appeared to be fixed properly ... (e.g., Figs. 1C, 2B) - I don't see any mitochondria in any figures. Perhaps this observation should be noted as "not shown"?

p.3 The images shown in Figs. 1, 2, 4 - The figures should be called out in the order; in this case, Fig 3 has not been called out yet.

p.4 we did not find any other connecting structures - Because the sample was processed by traditional EM methods, it's safer to add a caveat that other connecting structures could be missed if they were disrupted by sample prep or if they did not pick up stain as well as the two structures presented in this paper.

p.7 we expect such structures to be commonly seen in other cell types as well if they are examined carefully - Instead of saying that examinations should be done "carefully", it would be more helpful to specify how other cell types should be examined. This work shows that the bridges can be found if the cells are either sectioned parallel to the spindle axis or if a sufficiently large volume is sampled.

Please use consistent spelling/hyphenation of ultrafine/ultra-fine and word choice (strands vs. bridges).

****Referees cross-commenting****

I agree with my co-reviewers's comments and have no further suggestions.

This may be the first use of electron tomography to study the structural details of tethers that connect chromosomes in anaphase cells. The data is of sufficient quality to reveal differences in density. Namely, one class of tether appears to be an extension of the chromosome while the other class is composed of thin filaments. This study is novel in that it characterizes a mitosis-associated complex that is poorly studied compared to the microtubule-based spindle apparatus and the kinetochore. Hopefully, the tethers will draw more attention and further characterization by methods like super-resolution microscopy and cryo-electron microscopy. My expertise is in chromatin, mitotic machines, and cryo-electron tomography.

Review #3

****Summary:****

Tethers between telomeres of chromosomes in anaphase were inferred from earlier studies of laser microbeam cutting experiments. The current paper presents images from electron tomography of crane fly spermatocytes that substantiates the earlier inference. The authors deduce that the darker filaments and the lighter filaments that they visualize may be the structural tethers at telomeres.

****Major comments:****

The experiments are carefully done, and the conclusions are appropriately worded to qualify any caveats. This short communication is well-presented, and I have only a few comments. The authors should expand their list of references on bridges to include those listed by Warecki et al (Curr Biol 33:1-17, 2023; refs 15-26, etc). The authors present arguments that the tethers are not the DNA bridges observed by others. However, they should try to address this experimentally by treatment of their preparations with DNase to see if the thick and/or thin filaments disappear. Moreover, they should discuss in more detail the possible functions of (DNA) bridges, including the recent model from Bill Sullivan's lab (Warecki et al, Curr Biol, 2023) that they help to retain fragments of broken chromosomes. In addition, the authors should summarize the various proteins that may be associated with the bridges (as enumerated in the Warecki et al 2023 paper).

The authors could add a sentence to the Results or Discussion of whether the thicker tethers might become stretched as anaphase progresses to become the thinner tethers (Fig. 4G).

The authors may want to add a few sentences to the Discussion about the "chromosomal bouquet" stage of leptotene of meiosis prophase I where the telomeres of chromosomes seem pulled together and associate with the nuclear envelope --- they could speculate if this might also be due to the tethers that they describe in spermatocytes.

****Minor comments:****

A few additional comments are as follows:

p. 2 last sentence of first paragraph -modify the wording about "no structural evidence that identifies physical connections between separating telomeres", since there is some information from genetic and cell biology light microscopy experiments. Perhaps simply change "structural" to "ultrastructural".

p. 6, 5th line of second paragraph - change "ribosome DNA" to "ribosomal DNA"

Figure 1D - add the chromosome to the right of the schematic model (as suggested by Fig. 1B).

p. 17 (Methods), line 10 of first paragraph - state if this is light or heavy Halocarbon oil (give details).

p. 17 (Methods), line 12 of first paragraph- state the concentration for fibrinogen and for thrombin.

p. 17 (Methods), line 4 of second paragraph - is there any data to show that the filaments (tethers) occur if there is no cold shock?

****Referees cross-commenting****

I concur with Reviewers #1 and #2 that this is a fine paper that should be published. My detailed comments submitted with my review are simply meant as revisions to further strengthen this paper.

Strengths: This is an important conceptual advance and the carefully done ultrastructural imaging provides the foundation for future studies that could delve into the molecular composition and functional significance of the tethers at telomeres of anaphase chromosomes seen here by 3D electron microscopy.

Limitations: the molecular composition and functional roles are not yet known for the tethers seen here by 3D

electron microscopy, but to do so would involve an entire new program of experimentation.

Advances: there have only been two earlier ultrastructural papers on tethers at telomeres, and the tethers were peripheral to the main focus of those papers. Thus, the current paper extends our ultrastructural information about tethers.

Audience: this work is of importance for scientists who study the mechanics of chromosome movement on spindles, including regulation to combat aneuploidy. This work will also be important for a broader audience to inform them about transmission of the hereditary information to daughter cells.

1. General Statements [optional]

Our manuscript has been reviewed by three reviewers. As it can be seen in the reviewers' comments below, all the reviewers appreciated the clarity, quality, and significance of our work, and supported its publication. The reviewers suggested several points to improve our manuscript, which we have addressed by showing additional data and modifying the text, as outlined in detail below.

This section is mandatory. Please insert a point-by-point reply describing the revisions that were already carried out and included in the transferred manuscript.

Point-by-point response (referee comments in **blue italics**, our response in black)

Reviewer #1 (Evidence, reproducibility and clarity (Required)):

Summary: Forer and Otsuka provide first-rate evidence for tethers fixed in place between separating anaphase chromosomes using electron tomography. The authors traced the anaphase movement of a number of living cells before fixation for examination using electron tomography. The manuscript is clearly written and provides an excellent introduction and discussion of the known literature. The reader will have an excellent background to see the importance of this work.

Major comments:

- Are the claims and the conclusions supported by the data or do they require additional experiments or analyses to support them?

No further experiments are needed. The data are very supportive, and extremely clear.

- Are the data and the methods presented in such a way that they can be reproduced? Yes.

- Are the experiments adequately replicated and statistical analysis adequate? Yes.

Minor comments:

- Are prior studies referenced appropriately? Yes.*
- Are the text and figures clear and accurate? Absolutely.*
- Do you have suggestions that would help the authors improve the presentation of their data and conclusions?*

The authors are to be congratulated on their major contribution to this study on tethers between separated daughter chromosomes. It is a tour de force to go from the living cells to fixing and identifying the same separated chromosomes using electron tomography to see the ultrastructure of the fibers seen in vivo.

Referees cross-commenting

Thank you reviewer #2. The manuscript should be published. It is an excellent contribution.

We thank the reviewer for the appreciation of the clarity and quality of our work.

Reviewer #1 (Significance (Required)):

Provide contextual information to readers (editors and researchers) about the novelty of the study, its value for the field and the communities that might be interested.

This manuscript is the first to use electron tomography to identify the tethers between separated anaphase chromosomes. Forer and the late Michael Berns and their co-authors have published a number of papers using phase microscopy and lasers to report on the physical nature and elastic properties of these fibers in the past. Forer and Otsuka have presented first-rate evidence for the reality of these structures using electron tomography. This manuscript should be highlighted in the published journal. The chemical identity of these fibers as the authors state is unclear.

The following aspects are important:

- Audience: describe the type of audience ("specialized", "broad", "basic research", "translational/clinical", etc...) that will be interested or influenced by this research; how will this research be used by others; will it be of interest beyond the specific field?*

This exciting contribution will be read by anyone interested in mitosis. It will be of interest to all Cell Biologists because of the careful manner in which the living cells were studied before they were fixed for examination using electron tomography. The readers will be dreaming how they can use this process on their Cell Biology problems.

- Please define your field of expertise with a few keywords to help the authors contextualize your point of view. Indicate if there are any parts of the paper that you do not have sufficient expertise to evaluate.*

I am a cell Biologist who has made contributions, both in light microscopy and in transmission

microscopy on diving cells, both in tissue culture and in situ in avian and zebrafish embryos.

We thank the reviewer for appreciating the significance of our work.

Reviewer #2 (Evidence, reproducibility and clarity (Required)):

In this paper, the authors use light microscopy and electron tomography to study anaphase chromosomes in crane fly spermatocytes. They find that there are two "tether" structures that connect telomeres of sister chromatids. One tether is thicker (denser) and extends between sister chromatids during early but not late anaphase, whereas a second, less-dense tether maintains contact with both sister chromatids in all examined stages of anaphase. The paper makes arguments as to what the tethers could or could not be. Specifically, they are too numerous to be ultrafine DNA bridges seen in various normal or abnormal segregation events and they also do not affect anaphase chromosome motion the same way ultrafine DNA bridges do.

Major comments:

The major claim that there are tethers that connect sister chromatids in anaphase is supported by the data. Moreover, the data resolves two types of tethers on the basis of their density. While it is unclear what the composition of the tethers are, the paper makes a convincing case that they cannot be the DNA ultrafine bridges seen in other studies. The discussion has sufficient caveats that most readers will see that more work is needed to identify the composition of the two tethers. In my opinion, no further experiments are needed to support the modest claims of this paper. Therefore, I only have minor comments that may hopefully improve the paper's clarity.

We thank the reviewer for the positive evaluation of our work.

Minor comments:

It was argued that the tethers reported here were also seen in other species and cellular contexts, where the imaging work was done with projection EM imaging. Presumably, what is new here is the usage of electron tomography. It would help readers if the authors explained why the electron tomography done here was essential to arrive at key conclusions.

Thank you for the useful comment. We have added the explanation of why electron tomography was critical to visualise small tether structures to the last paragraph of the Discussion on page 7.

p.3 mitochondria appeared to be fixed properly ... (e.g., Figs. 1C, 2B) - I don't see any mitochondria in any figures. Perhaps this observation should be noted as "not shown"?

We thank the reviewer for pointing this out. We have added an electron micrograph of mitochondria to the Supplementary Figure 1.

p.3 The images shown in Figs. 1, 2, 4 - The figures should be called out in the order; in this case, Fig 3

has not been called out yet.

We have corrected the order of the figures.

p.4 we did not find any other connecting structures - Because the sample was processed by traditional EM methods, it's safer to add a caveat that other connecting structures could be missed if they were disrupted by sample prep or if they did not pick up stain as well as the two structures presented in this paper.

We have clarified that our sample was chemically fixed in the first paragraph of the Discussion on page 4. Because the details of how our samples were prepared are described in the Method section, we did not add further details to this paragraph.

p.7 we expect such structures to be commonly seen in other cell types as well if they are examined carefully - Instead of saying that examinations should be done "carefully", it would be more helpful to specify how other cell types should be examined. This work shows that the bridges can be found if the cells are either sectioned parallel to the spindle axis or if a sufficiently large volume is sampled.

We have now clarified that 3D electron microscopy techniques such as electron tomography are critical to visualise small tether structures in the last paragraph of the Discussion on page 7.

Please use consistent spelling/hyphenation of ultrafine/ultra-fine and word choice (strands vs. bridges).

Referees cross-commenting

I agree with my co-reviewers's comments and have no further suggestions.

Reviewer #2 (Significance (Required)):

This may be the first use of electron tomography to study the structural details of tethers that connect chromosomes in anaphase cells. The data is of sufficient quality to reveal differences in density. Namely, one class of tether appears to be an extension of the chromosome while the other class is composed of thin filaments. This study is novel in that it characterizes a mitosis-associated complex that is poorly studied compared to the microtubule-based spindle apparatus and the kinetochore. Hopefully, the tethers will draw more attention and further characterization by methods like super-resolution microscopy and cryo-electron microscopy. My expertise is in chromatin, mitotic machines, and cryo-electron tomography. We thank the reviewer for appreciating the novelty and the impact of our work.

Reviewer #3 (Evidence, reproducibility and clarity (Required)):

Summary:

Tethers between telomeres of chromosomes in anaphase were inferred from earlier studies of laser microbeam cutting experiments. The current paper presents images from electron tomography of crane fly spermatocytes that substantiates the earlier inference. The authors deduce that the darker filaments and the lighter filaments that they visualize may be the structural tethers at telomeres.

Major comments:

The experiments are carefully done, and the conclusions are appropriately worded to qualify any caveats. This short communication is well-presented, and I have only a few comments.

We thank the reviewer for appreciating the clarity and quality of our work.

The authors should expand their list of references on bridges to include those listed by Warecki et al (Curr Biol 33:1-17, 2023; refs 15-26, etc).

We do not think it is necessary to expand the list of references for ultra-fine DNA bridges. In the article we submitted, we discussed the Warecki et al article in the penultimate paragraph of the Discussion; we concluded that the bridges that Warecki et al described are different from ours in having so few per cell that they couldn't be tethers, and further that there was no evidence that those bridges were elastic. For those reasons, we do not find discussion of those proteins relevant to tethers, any more than would listing all the proteins associated with ultra-fine DNA bridges be relevant to the elastic tethers.

In the Discussion, we discussed data suggesting that a known elastic protein titin was present; that is as far as we wanted to go on speculation of what the elastic component of tethers might be.

The authors present arguments that the tethers are not the DNA bridges observed by others. However, they should try to address this experimentally by treatment of their preparations with DNase to see if the thick and/or thin filaments disappear.

While we agree that it would be important to identify the components of the tethers, we are concerned that those experiments are beyond the scope of this manuscript. Nevertheless, we appreciate the constructive suggestion for the future research direction.

Moreover, they should discuss in more detail the possible functions of (DNA) bridges, including the recent model from Bill Sullivan's lab (Warecki et al, Curr Biol, 2023) that they help to retain fragments of broken chromosomes. In addition, the authors should summarize the various proteins that may be associated with the bridges (as enumerated in the Warecki et al 2023 paper).

As we describe above, we concluded that the bridges Warecki et al described are different from the tethers that we report in our manuscript. Therefore, we do not think it is necessary to expand the discussion on the proteins and functions associated with ultra-fine DNA.

The authors could add a sentence to the Results or Discussion of whether the thicker tethers might become stretched as anaphase progresses to become the thinner tethers (Fig. 4G).

We thank the reviewer for this suggestion. We actually mentioned this possibility in the third paragraph of our Discussion on page 7.

The authors may want to add a few sentences to the Discussion about the "chromosomal bouquet" stage of leptotene of meiosis prophase I where the telomeres of chromosomes seem pulled together and associate with the nuclear envelope --- they could speculate if this might also be due to the tethers that they describe in spermatocytes.

This is a very interesting possibility. While we would refrain from adding this speculation to our manuscript as it is beyond the scope of the main points, it is certainly an interesting avenue of future research.

Minor comments:

A few additional comments are as follows:

p. 2 last sentence of first paragraph -modify the wording about "no structural evidence that identifies physical connections between separating telomeres", since there is some information from genetic and cell biology light microscopy experiments. Perhaps simply change "structural" to "ultrastructural".

We have changed the wording as the reviewer recommended.

p. 6, 5th line of second paragraph - change "ribosome DNA" to "ribosomal DNA"

We have corrected it.

Figure 1D - add the chromosome to the right of the schematic model (as suggested by Fig. 1B).

We are sorry for the confusion. In Figure 1D, the left half of the tethers are 3D modelled and shown. We have clarified this point by modifying the legend of Figure 1D.

p. 17 (Methods), line 10 of first paragraph - state if this is light or heavy Halocarbon oil (give details).

It is a mixture of heavy and light Halocarbon oil. We have clarified it on page 17.

p. 17 (Methods), line 12 of first paragraph- state the concentration for fibrinogen and for thrombin.

As we wrote in the original manuscript, the procedures are described in detail in our previous publication (Forer A. & Pickett-Heaps J. (2005) Fibrin clots keep non-adhering living cells in place on glass for perfusion or fixation. *Cell Biology International* **29**: 721–730). Nonetheless, to clarify this point, we have modified the text on page 17.

p. 17 (Methods), line 4 of second paragraph - is there any data to show that the filaments (tethers) occur if there is no cold shock?

Yes, we do see similar filamentous structures in the sample without cold shock. For your information, we show one of the electron micrographs below. In our manuscript, we show the data from the samples prepared with cold shock, because it better visualizes the filamentous structures. We now show these electron micrographs in the Supplementary Figure 2.

****Referees cross-commenting****

I concur with Reviewers #1 and #2 that this is a fine paper that should be published. My detailed comments submitted with my review are simply meant as revisions to further strengthen this paper. We thank the reviewer for supporting the publication of our manuscript.

Reviewer #3 (Significance (Required)):

Significance:

Strengths: This is an important conceptual advance and the carefully done ultrastructural imaging provides the foundation for future studies that could delve into the molecular composition and functional significance of the tethers at telomeres of anaphase chromosomes seen here by 3D electron microscopy.

Limitations: the molecular composition and functional roles are not yet known for the tethers seen here by 3D electron microscopy, but to do so would involve an entire new program of experimentation.

Advances: there have only been two earlier ultrastructural papers on tethers at telomeres, and the tethers were peripheral to the main focus of those papers. Thus, the current paper extends our ultrastructural information about tethers.

Audience: this work is of importance for scientists who study the mechanics of chromosome movement on spindles, including regulation to combat aneuploidy. This work will also be important for a broader audience to inform them about transmission of the hereditary information to daughter cells.

We thank the reviewer for appreciating the significance and the impact of our work.

August 4, 2023

RE: Life Science Alliance Manuscript #LSA-2023-02303

Dr. Shotaro Otsuka
Max Perutz Lab, Vienna
Austria

Dear Dr. Otsuka,

Thank you for submitting your revised manuscript entitled "Structural evidence for elastic tethers connecting separating chromosomes in crane-fly spermatocytes". We would be happy to publish your paper in Life Science Alliance pending final revisions necessary to meet our formatting guidelines.

- please upload all figure files as individual ones, including the supplementary figure files; all figure legends should only appear in the main manuscript file
- please add a Running Title and a Summary Blurb/Alternate Abstract to our system
- please add a Category for your manuscript in our system
- please add the Twitter handle of your host institute/organization as well as your own or/and one of the authors in our system
- please remove your figures from the main manuscript file
- please consult our manuscript preparation guidelines <https://www.life-science-alliance.org/manuscript-prep> and make sure your manuscript sections are in the correct order
- please add your main, supplementary figure, table, and movie legends to the main manuscript text after the references section;
- please add authors' contributions to our system as well
- please add a conflict of interest statement to your main manuscript text
- please add callouts for Figures 1D, 3A, C; S2A and B to your main manuscript text
- you may consider uploading Figure 5 as a Graphical Abstract rather than as a figure, but this is up to you

A. FINAL FILES:

B. MANUSCRIPT ORGANIZATION AND FORMATTING:

Sincerely,

August 7, 2023

RE: Life Science Alliance Manuscript #LSA-2023-02303R

Dr. Shotaro Otsuka
Max Perutz Labs
Dr. Bohr Gasse 9
Vienna 1030
Austria

Dear Dr. Otsuka,

Thank you for submitting your Research Article entitled "Structural evidence for elastic tethers connecting separating chromosomes in crane-fly spermatocytes". It is a pleasure to let you know that your manuscript is now accepted for publication in Life Science Alliance. Congratulations on this interesting work.

DISTRIBUTION OF MATERIALS:

Again, congratulations on a very nice paper. I hope you found the review process to be constructive and are pleased with how the manuscript was handled editorially. We look forward to future exciting submissions from your lab.

Sincerely,
